# Proximal SCOPE for Distributed Sparse Learning

**Shen-Yi Zhao**
National Key Lab. for Novel Software Tech.
Dept. of Comp. Sci. and Tech.
Nanjing University, Nanjing 210023, China
zhaosy@lamda.nju.edu.cn

**Gong-Duo Zhang**
National Key Lab. for Novel Software Tech.
Dept. of Comp. Sci. and Tech.
Nanjing University, Nanjing 210023, China
zhanggd@lamda.nju.edu.cn

**Ming-Wei Li**
National Key Lab. for Novel Software Tech.
Dept. of Comp. Sci. and Tech.
Nanjing University, Nanjing 210023, China
limw@lamda.nju.edu.cn

**Wu-Jun Li**
National Key Lab. for Novel Software Tech.
Dept. of Comp. Sci. and Tech.
Nanjing University, Nanjing 210023, China
liwujun@nju.edu.cn

## Abstract

Distributed sparse learning with a cluster of multiple machines has attracted much attention in machine learning, especially for large-scale applications with high-dimensional data. One popular way to implement sparse learning is to use $L_1$ regularization. In this paper, we propose a novel method, called proximal SCOPE (pSCOPE), for distributed sparse learning with $L_1$ regularization. pSCOPE is based on a cooperative autonomous local learning (CALL) framework. In the CALL framework of pSCOPE, we find that the data partition affects the convergence of the learning procedure, and subsequently we define a metric to measure the goodness of a data partition. Based on the defined metric, we theoretically prove that pSCOPE is convergent with a linear convergence rate if the data partition is good enough. We also prove that better data partition implies faster convergence rate. Furthermore, pSCOPE is also communication efficient. Experimental results on real data sets show that pSCOPE can outperform other state-of-the-art distributed methods for sparse learning.

## 1 Introduction

Many machine learning models can be formulated as the following regularized empirical risk minimization problem:

$$\min_{\mathbf{w} \in \mathbb{R}^d} \ P(\mathbf{w}) = \frac{1}{n} \sum_{i=1}^{n} f_i(\mathbf{w}) + R(\mathbf{w}), \tag{1}$$

where $\mathbf{w}$ is the parameter to learn, $f_i(\mathbf{w})$ is the loss on training instance $i$, $n$ is the number of training instances, and $R(\mathbf{w})$ is a regularization term. Recently, sparse learning, which tries to learn a sparse model for prediction, has become a hot topic in machine learning. There are different ways to implement sparse learning [28, 30]. One popular way is to use $L_1$ regularization, i.e., $R(\mathbf{w}) = \lambda \|\mathbf{w}\|_1$. In this paper, we focus on sparse learning with $R(\mathbf{w}) = \lambda \|\mathbf{w}\|_1$. Hence, in the following content of this paper, $R(\mathbf{w}) = \lambda \|\mathbf{w}\|_1$ unless otherwise stated.

One traditional method to solve (1) is proximal gradient descent (pGD) [2], which can be written as follows:

$$\mathbf{w}_{t+1} = \text{prox}_{R,\eta}(\mathbf{w}_t - \eta \nabla F(\mathbf{w}_t)), \tag{2}$$

where $F(\mathbf{w}) = \frac{1}{n}\sum_{i=1}^{n} f_i(\mathbf{w})$, $\mathbf{w}_t$ is the value of $\mathbf{w}$ at iteration $t$, $\eta$ is the learning rate, $prox$ is the proximal mapping defined as

$$\text{prox}_{R,\eta}(\mathbf{u}) = \arg\min_{\mathbf{v}} (R(\mathbf{v}) + \frac{1}{2\eta}\|\mathbf{v} - \mathbf{u}\|^2). \tag{3}$$

Recently, stochastic learning methods, including stochastic gradient descent (SGD) [18], stochastic average gradient (SAG) [22], stochastic variance reduced gradient (SVRG) [10], and stochastic dual coordinate ascent (SDCA) [24], have been proposed to speedup the learning procedure in machine learning. Inspired by the success of these stochastic learning methods, proximal stochastic methods, including proximal SGD (pSGD) [11, 6, 26, 4], proximal block coordinate descent (pBCD) [29, 31, 21], proximal SVRG (pSVRG) [32] and proximal SDCA (pSDCA) [25], have also been proposed for sparse learning in recent years. All these proximal stochastic methods are sequential (serial) and implemented with one single thread.

The serial proximal stochastic methods may not be efficient enough for solving large-scale sparse learning problems. Furthermore, the training set might be distributively stored on a cluster of multiple machines in some applications. Hence, distributed sparse learning [1] with a cluster of multiple machines has attracted much attention in recent years, especially for large-scale applications with high-dimensional data. In particular, researchers have recently proposed several distributed proximal stochastic methods for sparse learning [15, 17, 13, 16, 27][1].

One main branch of the distributed proximal stochastic methods includes distributed pSGD (dpSGD) [15], distributed pSVRG (dpSVRG) [9, 17] and distributed SVRG (DSVRG) [13]. Both dpSGD and dpSVRG adopt a centralized framework and mini-batch based strategy for distributed learning. One typical implementation of a centralized framework is based on Parameter Server [14, 33], which supports both synchronous and asynchronous communication strategies. One shortcoming of dpSGD and dpSVRG is that the communication cost is high. More specifically, the communication cost of each epoch is $O(n)$, where $n$ is the number of training instances. DSVRG adopts a decentralized framework with lower communication cost than dpSGD and dpSVRG. However, in DSVRG only one worker is updating parameters locally and all other workers are idling at the same time.

Another branch of the distributed proximal stochastic methods is based on block coordinate descent [3, 20, 7, 16]. Although in each iteration these methods update only a block of coordinates, they usually have to pass through the whole data set. Due to the partition of data, it also brings high communication cost in each iteration.

Another branch of the distributed proximal stochastic methods is based on SDCA. One representative is PROXCOCOA+ [27]. Although PROXCOCOA+ has been theoretically proved to have a linear convergence rate with low communication cost, we find that it is not efficient enough in experiments.

In this paper, we propose a novel method, called proximal SCOPE (pSCOPE), for distributed sparse learning with $L_1$ regularization. pSCOPE is a proximal generalization of the scalable composite optimization for learning (SCOPE) [34]. SCOPE cannot be used for sparse learning, while pSCOPE can be used for sparse learning. The contributions of pSCOPE are briefly summarized as follows:

- pSCOPE is based on a cooperative autonomous local learning (CALL) framework. In the CALL framework, each worker in the cluster performs autonomous local learning based on the data assigned to that worker, and the whole learning task is completed by all workers in a cooperative way. The CALL framework is communication efficient because there is no communication during the inner iterations of each epoch.

- pSCOPE is theoretically guaranteed to be convergent with a linear convergence rate if the data partition is good enough, and better data partition implies faster convergence rate. Hence, pSCOPE is also computation efficient.

- In pSCOPE, a *recovery strategy* is proposed to reduce the cost of proximal mapping when handling high dimensional sparse data.

- Experimental results on real data sets show that pSCOPE can outperform other state-of-the-art distributed methods for sparse learning.

## 2 Preliminary

In this paper, we use $\|\cdot\|$ to denote the $L_2$ norm $\|\cdot\|_2$, $\mathbf{w}^*$ to denote the optimal solution of (1). For a vector $\mathbf{a}$, we use $a^{(j)}$ to denote the $j$th coordinate value of $\mathbf{a}$. $[n]$ denotes the set $\{1, 2, \ldots, n\}$. For a function $h(\mathbf{a}; \mathbf{b})$, we use $\nabla h(\mathbf{a}; \mathbf{b})$ to denote the gradient of $h(\mathbf{a}; \mathbf{b})$ with respect to (w.r.t.) the first argument $\mathbf{a}$. Furthermore, we give the following definitions.

**Definition 1** *We call a function $h(\cdot)$ is L-smooth if it is differentiable and there exists a positive constant $L$ such that $\forall \mathbf{a}, \mathbf{b} : h(\mathbf{b}) \leq h(\mathbf{a}) + \nabla h(\mathbf{a})^T (\mathbf{b} - \mathbf{a}) + \frac{L}{2} \|\mathbf{a} - \mathbf{b}\|^2$.*

**Definition 2** *We call a function $h(\cdot)$ is convex if there exists a constant $\mu \geq 0$ such that $\forall \mathbf{a}, \mathbf{b} : h(\mathbf{b}) \geq h(\mathbf{a}) + \zeta^T (\mathbf{b} - \mathbf{a}) + \frac{\mu}{2} \|\mathbf{a} - \mathbf{b}\|^2$, where $\zeta \in \partial h(\mathbf{a}) = \{\mathbf{c} | h(\mathbf{b}) \geq h(\mathbf{a}) + \mathbf{c}^T (\mathbf{b} - \mathbf{a}), \forall \mathbf{a}, \mathbf{b}\}$. If $h(\cdot)$ is differentiable, then $\zeta = \nabla h(\mathbf{a})$. If $\mu > 0$, $h(\cdot)$ is called $\mu$-strongly convex.*

Throughout this paper, we assume that $R(\mathbf{w})$ is convex, $F(\mathbf{w}) = \frac{1}{n} \sum_{i=1}^{n} f_i(\mathbf{w})$ is strongly convex and each $f_i(\mathbf{w})$ is smooth. We do not assume that each $f_i(\mathbf{w})$ is convex.

## 3 Proximal SCOPE

In this paper, we focus on distributed learning with one master (server) and $p$ workers in the cluster, although the algorithm and theory of this paper can also be easily extended to cases with multiple servers like the Parameter Server framework [14, 33].

The parameter $\mathbf{w}$ is stored in the master, and the training set $D = \{\mathbf{x}_i, y_i\}_{i=1}^n$ are partitioned into $p$ parts denoted as $D_1, D_2, \ldots, D_p$. Here, $D_k$ contains a subset of instances from $D$, and $D_k$ will be assigned to the $k$th worker. $D = \bigcup_{k=1}^{p} D_k$. Based on this data partition scheme, the proximal SCOPE (pSCOPE) for distributed sparse learning is presented in Algorithm 1. The main task of master is to add and average vectors received from workers. Specifically, it needs to calculate the full gradient $\mathbf{z} = \nabla F(\mathbf{w}_t) = \frac{1}{n} \sum_{k=1}^{p} \mathbf{z}_k$. Then it needs to calculate $\mathbf{w}_{t+1} = \frac{1}{p} \sum_{k=1}^{p} \mathbf{u}_{k,M}$. The main task of workers is to update the local parameters $\mathbf{u}_{1,m}, \mathbf{u}_{2,m}, \ldots, \mathbf{u}_{p,m}$ initialized with $\mathbf{u}_{k,0} = \mathbf{w}_t$. Specifically, for each worker $k$, after it gets the full gradient $\mathbf{z}$ from master, it calculates a stochastic gradient

$$\mathbf{v}_{k,m} = \nabla f_{i_{k,m}}(\mathbf{u}_{k,m}) - \nabla f_{i_{k,m}}(\mathbf{w}_t) + \mathbf{z}, \tag{4}$$

and then update its local parameter $\mathbf{u}_{k,m}$ by a proximal mapping with learning rate $\eta$:

$$\mathbf{u}_{k,m+1} = \text{prox}_{R,\eta}(\mathbf{u}_{k,m} - \eta \mathbf{v}_{k,m}). \tag{5}$$

From Algorithm 1, we can find that pSCOPE is based on a cooperative autonomous local learning (CALL) framework. In the CALL framework, each worker in the cluster performs autonomous local learning based on the data assigned to that worker, and the whole learning task is completed by all workers in a cooperative way. The cooperative operation is mainly adding and averaging in the master. During the autonomous local learning procedure in each outer iteration which contains $M$ inner iterations (see Algorithm 1), there is no communication. Hence, the communication cost for each epoch of pSCOPE is constant, which is much less than the mini-batch based strategy with $O(n)$ communication cost for each epoch [15, 9, 17].

pSCOPE is a proximal generalization of SCOPE [34]. Although pSCOPE is mainly motivated by sparse learning with $L_1$ regularization, the algorithm and theory of pSCOPE can also be used for smooth regularization like $L_2$ regularization. Furthermore, when the data partition is good enough, pSCOPE can avoid the extra term $c(\mathbf{u}_{k,m} - \mathbf{w}_t)$ in the update rule of SCOPE, which is necessary for convergence guarantee of SCOPE.

## 4 Effect of Data Partition

In our experiment, we find that the data partition affects the convergence of the learning procedure. Hence, in this section we propose a metric to measure the goodness of a data partition, based on which the convergence of pSCOPE can be theoretically proved. Due to space limitation, the detailed proof of Lemmas and Theorems are moved to the long version [35].

---

**Algorithm 1** Proximal SCOPE

---
1: Initialize $\mathbf{w}_0$ and the learning rate $\eta$;
2: **Task of master**:
3: **for** $t = 0, 1, 2, ...T - 1$ **do**
4:     Send $\mathbf{w}_t$ to each worker;
5:     Wait until receiving $\mathbf{z}_1, \mathbf{z}_2, \ldots, \mathbf{z}_p$ from all workers;
6:     Calculate the full gradient $\mathbf{z} = \frac{1}{n} \sum_{k=1}^{p} \mathbf{z}_k$ and send $\mathbf{z}$ to each worker;
7:     Wait until receiving $\mathbf{u}_{1,M}, \mathbf{u}_{2,M}, \ldots, \mathbf{u}_{p,M}$ from all workers and calculate $\mathbf{w}_{t+1} = \frac{1}{p} \sum_{k=1}^{p} \mathbf{u}_{k,M}$;
8: **end for**
9: **Task of the $k$th worker**:
10: **for** $t = 0, 1, 2, ...T - 1$ **do**
11:     Wait until receiving $\mathbf{w}_t$ from master;
12:     Let $\mathbf{u}_{k,0} = \mathbf{w}_t$, calculate $\mathbf{z}_k = \sum_{i \in D_k} f_i(\mathbf{w}_t)$ and send $\mathbf{z}_k$ to master;
13:     Wait until receiving $\mathbf{z}$ from master;
14:     **for** $m = 0, 1, 2, ...M - 1$ **do**
15:         Randomly choose an instance $\mathbf{x}_{i_{k,m}} \in D_k$;
16:         Calculate $\mathbf{v}_{k,m} = \nabla f_{i_{k,m}}(\mathbf{u}_{k,m}) - \nabla f_{i_{k,m}}(\mathbf{w}_t) + \mathbf{z}$;
17:         Update $\mathbf{u}_{k,m+1} = \text{prox}_{R,\eta}(\mathbf{u}_{k,m} - \eta \mathbf{v}_{k,m})$;
18:     **end for**
19:     Send $\mathbf{u}_{k,M}$ to master
20: **end for**

---

## 4.1 Partition

First, we give the following definition:

**Definition 3** *Define* $\pi = [\phi_1(\cdot), \ldots, \phi_p(\cdot)]$. *We call* $\pi$ *a partition w.r.t.* $P(\cdot)$, *if* $F(\mathbf{w}) = \frac{1}{p} \sum_{k=1}^{p} \phi_k(\mathbf{w})$ *and each* $\phi_k(\cdot)$ $(k = 1, \ldots, p)$ *is* $\mu_k$-*strongly convex and* $L_k$-*smooth* $(\mu_k, L_k > 0)$. *Here,* $P(\cdot)$ *is defined in (1) and* $F(\cdot)$ *is defined in (2). We denote* $A(P) = \{\pi | \pi$ *is a partition w.r.t.* $P(\cdot)\}$.

**Remark 1** *Here,* $\pi$ *is an ordered sequence of functions. In particular, if we construct another partition* $\pi'$ *by permuting* $\phi_i(\cdot)$ *of* $\pi$, *we consider them to be two different partitions. Furthermore, two functions* $\phi_i(\cdot), \phi_j(\cdot)$ $(i \neq j)$ *in* $\pi$ *can be the same. Two partitions* $\pi_1 = [\phi_1(\cdot), \ldots, \phi_p(\cdot)]$, $\pi_2 = [\psi_1(\cdot), \ldots, \psi_p(\cdot)]$ *are considered to be equal, i.e.,* $\pi_1 = \pi_2$, *if and only if* $\phi_k(\mathbf{w}) = \psi_k(\mathbf{w})(k = 1, \ldots, p), \forall \mathbf{w}$.

For any partition $\pi = [\phi_1(\cdot), \ldots, \phi_p(\cdot)]$ w.r.t. $P(\cdot)$, we construct new functions $P_k(\cdot; \cdot)$ as follows:

$$P_k(\mathbf{w}; \mathbf{a}) = \phi_k(\mathbf{w}; \mathbf{a}) + R(\mathbf{w}), k = 1, \ldots, p \tag{6}$$

where $\phi_k(\mathbf{w}; \mathbf{a}) = \phi_k(\mathbf{w}) + G_k(\mathbf{a})^T \mathbf{w}$, $G_k(\mathbf{a}) = \nabla F(\mathbf{a}) - \nabla \phi_k(\mathbf{a})$, and $\mathbf{w}, \mathbf{a} \in \mathbb{R}^d$.

In particular, given a data partition $D_1, D_2, \ldots, D_p$ of the training set $D$, let $F_k(\mathbf{w}) = \frac{1}{|D_k|} \sum_{i \in D_k} f_i(\mathbf{w})$ which is also called the *local loss function*. Assume each $F_k(\cdot)$ is strongly convex and smooth, and $F(\mathbf{w}) = \frac{1}{p} \sum_{k=1}^{p} F_k(\mathbf{w})$. Then, we can find that $\pi = [F_1(\cdot), \ldots, F_p(\cdot)]$ is a partition w.r.t. $P(\cdot)$. By taking expectation on $\mathbf{v}_{k,m}$ defined in Algorithm 1, we obtain $\mathbb{E}[\mathbf{v}_{k,m} | \mathbf{u}_{k,m}] = \nabla F_k(\mathbf{u}_{k,m}) + G_k(\mathbf{w}_t)$. According to the theory in [32], in the inner iterations of pSCOPE, each worker tries to optimize the local objective function $P_k(\mathbf{w}; \mathbf{w}_t)$ using proximal SVRG with initialization $\mathbf{w} = \mathbf{w}_t$ and training data $D_k$, rather than optimizing $F_k(\mathbf{w}) + R(\mathbf{w})$. Then we call such a $P_k(\mathbf{w}; \mathbf{a})$ the *local objective function* w.r.t. $\pi$. Compared to the subproblem of PROXCOCOA+ (equation (2) in [27]), $P_k(\mathbf{w}; \mathbf{a})$ is more simple and there is no hyperparameter in it.

## 4.2 Good Partition

In general, the data distribution on each worker is different from the distribution of the whole training set. Hence, there exists a gap between each local optimal value and the global optimal value.

Intuitively, the whole learning algorithm has slow convergence rate or cannot even converge if this gap is too large.

**Definition 4** *For any partition $\pi$ w.r.t. $P(\cdot)$, we define the* Local-Global Gap *as*

$$l_\pi(\mathbf{a}) = P(\mathbf{w}^*) - \frac{1}{p} \sum_{k=1}^{p} P_k(\mathbf{w}_k^*(\mathbf{a}); \mathbf{a}),$$

*where* $\mathbf{w}_k^*(\mathbf{a}) = \arg\min_{\mathbf{w}} P_k(\mathbf{w}; \mathbf{a})$.

We have the following properties of Local-Global Gap:

**Lemma 1** $\forall \pi \in A(P)$, $l_\pi(\mathbf{a}) = P(\mathbf{w}^*) + \frac{1}{p} \sum_{k=1}^{p} H_k^*(-G_k(\mathbf{a})) \geq l_\pi(\mathbf{w}^*) = 0, \forall \mathbf{a}$, *where* $H_k^*(\cdot)$ *is the conjugate function of* $\phi_k(\cdot) + R(\cdot)$.

**Theorem 1** *Let* $R(\mathbf{w}) = \|\mathbf{w}\|_1$. $\forall \pi \in A(P)$, *there exists a constant* $\gamma < \infty$ *such that* $l_\pi(\mathbf{a}) \leq \gamma \|\mathbf{a} - \mathbf{w}^*\|^2, \forall \mathbf{a}$.

The result in Theorem 1 can be easily extended to smooth regularization which can be found in the long version [35].

According to Theorem 1, the local-global gap can be bounded by $\gamma \|\mathbf{a} - \mathbf{w}^*\|^2$. Given a specific $\mathbf{a}$, the smaller $\gamma$ is, the smaller the local-global gap will be. Since the constant $\gamma$ only depends on the partition $\pi$, intuitively $\gamma$ can be used to evaluate the *goodness* of a partition $\pi$. We define a *good partition* as follows:

**Definition 5** *We call $\pi$ a $(\epsilon, \xi)$-good partition w.r.t. $P(\cdot)$ if $\pi \in A(P)$ and*

$$\gamma(\pi; \epsilon) \triangleq \sup_{\|\mathbf{a}-\mathbf{w}^*\|^2 \geq \epsilon} \frac{l_\pi(\mathbf{a})}{\|\mathbf{a} - \mathbf{w}^*\|^2} \leq \xi. \tag{7}$$

In the following, we give the bound of $\gamma(\pi; \epsilon)$.

**Lemma 2** *Assume* $\pi = [F_1(\cdot), \ldots, F_p(\cdot)]$ *is a partition w.r.t. $P(\cdot)$, where $F_k(\mathbf{w}) = \frac{1}{|D_k|} \sum_{i \in D_k} f_i(\mathbf{w})$ is the local loss function, each $f_i(\cdot)$ is Lipschitz continuous with bounded domain and sampled from some unknown distribution $\mathbb{P}$. If we assign these $\{f_i(\cdot)\}$ uniformly to each worker, then with high probability, $\gamma(\pi; \epsilon) \leq \frac{1}{p} \sum_{k=1}^{p} \mathcal{O}(1/(\epsilon\sqrt{|D_k|}))$. Moreover, if $l_\pi(\mathbf{a})$ is convex w.r.t. $\mathbf{a}$, then $\gamma(\pi; \epsilon) \leq \frac{1}{p} \sum_{k=1}^{p} \mathcal{O}(1/\sqrt{\epsilon|D_k|})$. Here we ignore the* log *term and dimensionality d.*

For example, in Lasso regression, it is easy to get that the corresponding local-global gap $l_\pi(\mathbf{a})$ is convex according to Lemma 1 and the fact that $G_k(\mathbf{a})$ is an affine function in this case.

Lemma 2 implies that as long as the size of training data is large enough, $\gamma(\pi; \epsilon)$ will be small and $\pi$ will be a good partition. Please note that the *uniformly* here means each $f_i(\cdot)$ will be assigned to one of the $p$ workers and each worker has the equal probability to be assigned. We call the partition resulted from uniform assignment *uniform partition* in this paper. With uniform partition, each worker will have almost the same number of instances. As long as the size of training data is large enough, uniform partition is a good partition.

## 5  Convergence of Proximal SCOPE

In this section, we will prove the convergence of Algorithm 1 for proximal SCOPE (pSCOPE) using the results in Section 4.

**Theorem 2** *Assume $\pi = [F_1(\cdot), \ldots, F_p(\cdot)]$ is a $(\epsilon, \xi)$-good partition w.r.t. $P(\cdot)$. For convenience, we set $\mu_k = \mu, L_k = L, k = 1, 2 \ldots, p$. If $\|\mathbf{w}_t - \mathbf{w}^*\|^2 \geq \epsilon$, then*

$$\mathbb{E}\|\mathbf{w}_{t+1} - \mathbf{w}^*\|^2 \leq [(1 - \mu\eta + 2L^2\eta^2)^M + \frac{2L^2\eta + 2\xi}{\mu - 2L^2\eta}]\|\mathbf{w}_t - \mathbf{w}^*\|^2.$$

Because smaller $\xi$ means better partition and the partition $\pi$ corresponds to data partition in Algorithm 1, we can see that *better data partition implies faster convergence rate.*

**Corollary 1** *Assume $\pi = [F_1(\cdot), \ldots, F_p(\cdot)]$ is a $(\epsilon, \frac{\mu}{8})$-good partition w.r.t. $P(\cdot)$. For convenience, we set $\mu_k = \mu, L_k = L, k = 1, 2 \ldots, p$. If $\|\mathbf{w}_t - \mathbf{w}^*\|^2 \geq \epsilon$, taking $\eta = \frac{\mu}{12L^2}$, $M = 20\kappa^2$, where $\kappa = \frac{L}{\mu}$ is the conditional number, then we have $\mathbb{E}\|\mathbf{w}_{t+1} - \mathbf{w}^*\|^2 \leq \frac{3}{4}\|\mathbf{w}_t - \mathbf{w}^*\|^2$. To get the $\epsilon$-suboptimal solution, the computation complexity of each worker is $O((n/p + \kappa^2)\log(\frac{1}{\epsilon}))$.*

**Corollary 2** *When $p = 1$, which means we only use one worker, pSCOPE degenerates to proximal SVRG [32]. Assume $F(\cdot)$ is $\mu$-strongly convex ($\mu > 0$) and $L$-smooth. Taking $\eta = \frac{\mu}{6L^2}$, $M = 13\kappa^2$, we have $\mathbb{E}\|\mathbf{w}_{t+1} - \mathbf{w}^*\|^2 \leq \frac{3}{4}\|\mathbf{w}_t - \mathbf{w}^*\|^2$. To get the $\epsilon$-optimal solution, the computation complexity is $O((n + \kappa^2)\log(\frac{1}{\epsilon}))$.*

We can find that pSCOPE has a linear convergence rate if the partition is $(\epsilon, \xi)$-good, which implies pSCOPE is computation efficient and we need $T = O(\log(\frac{1}{\epsilon}))$ outer iterations to get a $\epsilon$-optimal solution. For all inner iterations, each worker updates $\mathbf{u}_{k,m}$ without any communication. Hence, the communication cost is $O(\log(\frac{1}{\epsilon}))$, which is much smaller than the mini-batch based strategy with $O(n)$ communication cost for each epoch [15, 9, 17].

Furthermore, in the above theorems and corollaries, we only assume that the local loss function $F_k(\cdot)$ is strongly convex. We do not need each $f_i(\cdot)$ to be convex. Hence, $M = O(\kappa^2)$ and it is weaker than the assumption in proximal SVRG [32] whose computation complexity is $O((n + \kappa)\log(\frac{1}{\epsilon}))$ when $p = 1$. In addition, without convexity assumption for each $f_i(\cdot)$, our result for the degenerate case $p = 1$ is consistent with that in [23].

# 6 Handle High Dimensional Sparse Data

For the cases with high dimensional sparse data, we propose *recovery strategy* to reduce the cost of proximal mapping so that it can accelerate the training procedure. Here, we adopt the widely used linear model with elastic net [36] as an example for illustration, which can be formulated as follows: $\min_{\mathbf{w}} P(\mathbf{w}) := \frac{1}{n}\sum_{i=1}^n h_i(\mathbf{x}_i^T\mathbf{w}) + \frac{\lambda_1}{2}\|\mathbf{w}\|^2 + \lambda_2\|\mathbf{w}\|_1$, where $h_i : \mathbb{R} \to \mathbb{R}$ is the loss function. We assume many instances in $\{\mathbf{x}_i \in \mathbb{R}^d | i \in [n]\}$ are sparse vectors and let $C_i = \{j | x_i^{(j)} \neq 0\}$.

Proximal mapping is unacceptable when the data dimensionality $d$ is too large, since we need to execute the conditional statements $O(Md)$ times which is time consuming. Other methods, like proximal SGD and proximal SVRG, also suffer from this problem.

Since $z^{(j)}$ is a constant during the update of local parameter $\mathbf{u}_{k,m}$, we will design a *recovery strategy* to recover it when necessary. More specifically, in each inner iteration, with the random index $s = i_{k,m}$, we only *recover* $u^{(j)}$ to calculate the inner product $\mathbf{x}_s^T\mathbf{u}_{k,m}$ and update $u_{k,m}^{(j)}$ for $j \in C_s$. For those $j \notin C_s$, we do not immediately update $u_{k,m}^{(j)}$. The basic idea of these recovery rules is: for some coordinate $j$, we can calculate $u_{k,m_2}^{(j)}$ directly from $u_{k,m_1}^{(j)}$, rather than doing iterations from $m = m_1$ to $m_2$. Here, $0 \leq m_1 < m_2 \leq M$. At the same time, the new algorithm is totally equivalent to Algorithm 1. It will save about $O(d(m_2 - m_1)(1 - \rho))$ times of conditional statements, where $\rho$ is the sparsity of $\{\mathbf{x}_i \in \mathbb{R}^d | i \in [n]\}$. This reduction of computation is significant especially for high dimensional sparse training data. Due to space limitation, the complete rules are moved to the long version [35]. Here we only give one case of our recovery rules in Lemma 3.

**Lemma 3** *(Recovery Rule) We define the sequence $\{\alpha_q\}$ as: $\alpha_0 = 0$ and for $q = 1, 2, \ldots, \alpha_q = \sum_{i=1}^q (1 - \lambda_1\eta)^{i-1}/(1 - \lambda_1\eta)^q$. For the coordinate $j$ and constants $m_1, m_2$, if $j \notin C_{i_{k,m}}$ for any $m \in [m_1, m_2 - 1]$. If $|z^{(j)}| < \lambda_2, u_{k,m_1}^{(j)} > 0$, then the relation between $u_{k,m_1}^{(j)}$ and $u_{k,m_2}^{(j)}$ can be summarized as follows: define $q_0$ which satisfies $\alpha_{q_0}\eta(z^{(j)} + \lambda_2) \leq u_{k,m_1}^{(j)} < \alpha_{q_0+1}\eta(z^{(j)} + \lambda_2)$,*

   *1. If $m_2 - m_1 \leq q_0$, then $u_{k,m_2}^{(j)} = (1 - \lambda_1\eta)^{m_2-m_1}[u_{k,m_1}^{(j)} - \alpha_{m_2-m_1}\eta(z^{(j)} + \lambda_2)]$.*

   *2. If $m_2 - m_1 > q_0$, then $u_{k,m_2}^{(j)} = 0$.*

# 7 Experiment

We use two sparse learning models for evaluation. One is logistic regression (LR) with elastic net [36]: $P(\mathbf{w}) = \frac{1}{n}\sum_{i=1}^{n}\log(1 + e^{-y_i\mathbf{x}_i^T\mathbf{w}}) + \frac{\lambda_1}{2}\|\mathbf{w}\|^2 + \lambda_2\|\mathbf{w}\|_1$. The other is Lasso regression [28]: $P(\mathbf{w}) = \frac{1}{2n}\sum_{i=1}^{n}(\mathbf{x}_i^T\mathbf{w} - y_i)^2 + \lambda_2\|\mathbf{w}\|_1$. All experiments are conducted on a cluster of multiple machines. The CPU for each machine has 12 Intel E5-2620 cores, and the memory of each machine is 96GB. The machines are connected by 10GB Ethernet. Evaluation is based on four datasets in Table 1: cov, rcv1, avazu, kdd2012. All of them can be downloaded from LibSVM website [2].

Table 1: Datasets

|  | #instances | #features | $\lambda_1$ | $\lambda_2$ |
|---|---|---|---|---|
| cov | 581,012 | 54 | $10^{-5}$ | $10^{-5}$ |
| rcv1 | 677,399 | 47,236 | $10^{-5}$ | $10^{-5}$ |
| avazu | 23,567,843 | 1,000,000 | $10^{-7}$ | $10^{-5}$ |
| kdd2012 | 119,705,032 | 54,686,452 | $10^{-8}$ | $10^{-5}$ |

## 7.1 Baselines

We compare our pSCOPE with six representative baselines: proximal gradient descent based method FISTA [2], ADMM type method DFAL [1], newton type method mOWL-QN [8], proximal SVRG based method AsyProx-SVRG [17], proximal SDCA based method PROXCOCOA+ [27], and distributed block coordinate descent DBCD [16]. FISTA and mOWL-QN are serial. We design distributed versions of them, in which workers distributively compute the gradients and then master gathers the gradients from workers for parameter update.

All methods use 8 workers. One master will be used if necessary. Unless otherwise stated, all methods except DBCD and PROXCOCOA+ use the same data partition, which is got by uniformly assigning each instance to each worker (uniform partition). Hence, different workers will have almost the same number of instances. This uniform partition strategy satisfies the condition in Lemma 2. Hence, it is a *good* partition. DBCD and PROXCOCOA+ adopt a coordinate distributed strategy to partition the data.

## 7.2 Results

The convergence results of LR with elastic net and Lasso regression are shown in Figure 1. DBCD is too slow, and hence we will separately report the time of it and pSCOPE when they get $10^{-3}$-suboptimal solution in Table 2. AsyProx-SVRG is slow on the two large datasets avazu and kdd2012, and hence we only present the results of it on the datasets cov and rcv1. From Figure 1 and Table 2, we can find that pSCOPE outperforms all the other baselines on all datasets.

Table 2: Time comparison (in second) between pSCOPE and DBCD.

|  |  | pSCOPE | DBCD |
|---|---|---|---|
| LR | cov | 0.32 | 822 |
|  | rcv1 | 3.78 | > 1000 |
| Lasso | cov | 0.06 | 81.9 |
|  | rcv1 | 3.09 | > 1000 |

## 7.3 Speedup

We also evaluate the speedup of pSCOPE on the four datasets for LR. We run pSCOPE and stop it when the gap $P(\mathbf{w}) - P(\mathbf{w}^*) \leq 10^{-6}$. The speedup is defined as: Speedup = (Time using one worker)/(Time using $p$ workers). We set $p = 1, 2, 4, 8$. The speedup results are in Figure 2 (a). We can find that pSCOPE gets promising speedup.

## 7.4 Effect of Data Partition

We evaluate pSCOPE under different data partitions. We use two datasets cov and rcv1 for illustration, since they are balanced datasets which means the number of positive instances is almost the same as

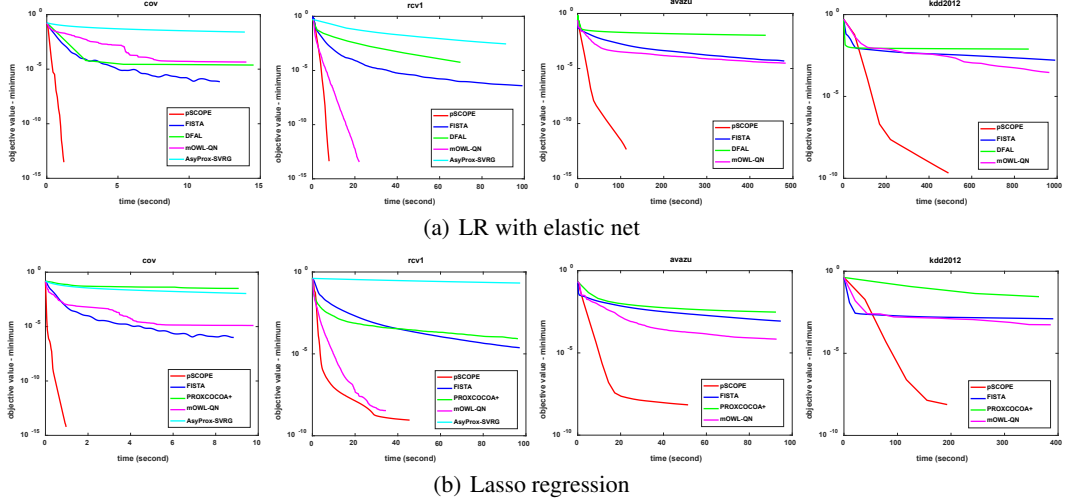

(a) LR with elastic net

(b) Lasso regression

Figure 1: Evaluation with baselines on two models.

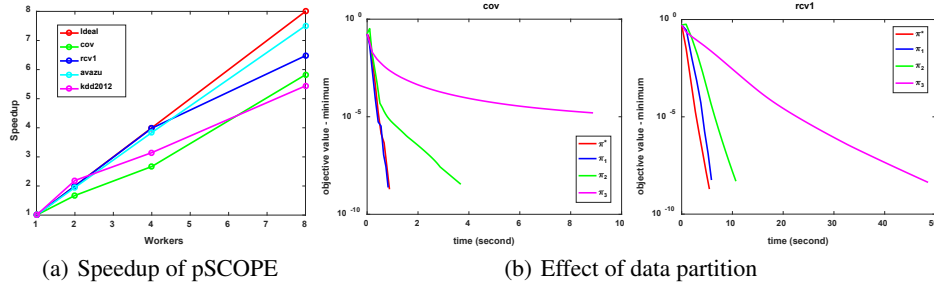

(a) Speedup of pSCOPE      (b) Effect of data partition

Figure 2: Speedup and effect of data partition

that of negative instances. For each dataset, we construct four data partitions: $\pi^*$ (each worker has the whole data), $\pi_1$ (uniform partition); $\pi_2$ (75% positive instances and 25% negative instances are on the first 4 workers, and other instances are on the last 4 workers), $\pi_3$ (all positive instances are on the first 4 workers, and all negative instances are on the last 4 workers).

The convergence results are shown in Figure 2 (b). We can see that data partition does affect the convergence of pSCOPE. The best partition $\pi^*$ achieves the best performance[3]. The performance of uniform partition $\pi_1$ is similar to that of the best partition $\pi^*$, and is better than the other two data partitions. In real applications with large-scale dataset, it is impractical to assign each worker the whole dataset. Hence, we prefer to choose uniform partition $\pi_1$ in real applications, which is also adopted in above experiments of this paper.

# 8 Conclusion

In this paper, we propose a novel method, called pSCOPE, for distributed sparse learning. Furthermore, we theoretically analyze how the data partition affects the convergence of pSCOPE. pSCOPE is both communication and computation efficient. Experiments on real data show that pSCOPE can outperform other state-of-the-art methods to achieve the best performance.

### Acknowledgements

This work is partially supported by the "DengFeng" project of Nanjing University.

## Footnotes

[1]In this paper, we mainly focus on distributed sparse learning with $L_1$ regularization. The distributed methods for non-sparse learning, like those in [19, 5, 12], are not considered.

[2]https://www.csie.ntu.edu.tw/~cjlin/libsvmtools/datasets/

[3]The proof that $\pi^*$ is the best partition can be found in the long version [35].

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
