[Reviews · NeurIPS 2018]

Reviewer 1



This paper mainly focused on solving regularized finite-sum problem in distributed setting. The first contribution is that it developed pSCOPE, which is an improvement over an existing SCOPE by combining proximal mapping to deal with non-smooth regularization term. Then a new measure to determine the quality of data partition is introduced, and the convergence of pSCOPE is analyzed with this measure. Furthermore, a trick for dealing with sparse data efficiently for pSCOPE is also proposed. Finally, some numerical comparisons were conducted to show the efficiency of pSCOPE method. I do not think turning an existing method SCOPE into the proximal version is quite interesting or has great contribution. Though the authors listed several differences between SCOPE and pSCOPE, I still think these differences are tiny. On the other hand, I personally prefer the part about how to measure the quality of data partition and how it affects convergence rate, which should be novel. However, it is pity that such theory was not be utilized to guide the data partition in the paper. The convergence rate is in fact poor, which has a quadratic dependence on the condition number. I do not agree this quadratic dependence is due to distributed computation. The effect of distribution should be reflected through $\xi$ instead of $\kappa$ (note that even for p=1, you still have $\kappa^2$). While most SVRG variants, no matter distributed or non-distributed, do have linear dependence on condition number. Some experiment results look strange for me. But the convergence behavior of the baselines in some figures are really weird. Some of them seem not convergent at all. I am not challenging authors’ experiment here. I sincerely suggest the authors can recheck their experiments carefully, especially the implementation of these baselines. There also exist several formatting issues in the paper. For example, the math equation should be more professional, especially the proximal mapping operator. And The figures are really too small for publishing. I believe in line 162 that the square of the norm is missing. Based on the above facts, I mainly remain neutral but slightly tent to accept this paper. *********************************************************************** I have read authors’ feedback. The explanation to why the convergence has quadratic dependence on condition number is reasonable now. However, I still prefer a bound with linear dependence on condition number, since in most applications we do have individual convexity. Hence, I do not plan to change the score.

Reviewer 2



The paper introduce a proximal algorithm for solving composite optimization problem with sparse l1 norm regularizer in a distributed fashion. It extends a previously introduced method call SCOPE (S-Y Zhao etal 2017) in order to handle sparse regularization. As a main contribution, the paper highlight the dependence of the partition of the data on the convergence rate (which is linear given strongly convex loss part). Numerical experiments demonstrate practical efficiency of the proposed method. As stated by the authors, the pSCOPE is limited to l1 norm regularizer whereas this restriction does not seem very legitimate. It would be interesting to keep a general framework that applies to any regularization R with a computable proximal mapping. According to (M. Jaggi, 2013) the Lasso and SVM are equivalent in an optimization point of view. While SCOPE applies to SVM, the authors claim that it does not apply to the lasso. So the authors should justify properly the assertions on the limitation of SCOPE. The assumptions on theoretical results are pretty confusing and sometimes does not apply for the examples given in the paper. For instance the Lipschitz continuous assumption on f_i in lemma 2 does not hold for the Lasso. Also the strong convexity assumptions on F_k in Theorem 2 does not hold if the ridge regularization is not added as in the elastic net case. So the convergence rate presented does not holds for the vanilla lasso and logistic regression with l1 regularizer. So the second point in line 73 should be clarified or weakened. M in line 106 is not defined in line 12 of algorithm 1, is there a missing nabla? The definition 5 of a partition seems independent to P, so why it should be wrt P()? Lemma 2 in line 180 eta is not defined in theorem 2. I am not an expert in distributed optimization and I can not evaluate the significance of the contribution. This seems to me to be a rather direct extension.

Reviewer 3



In this paper the authors propose a proximal version of SCOPE, that can be used for distributed sparse learning. The paper is well-organized in general, but I suggest to have your submission proof-read for English style and grammar issues. Also, the definition 1, of Lipschitz functions, could be left out, as well as the conjugate function. Just add some reference, or put it in the appendix, so you get more space to explain better some points. The figures are also quite small, and it's difficult to see. In the printed version, Figure 3 is impossible to read. The lemmas and theorems seem to be ok, and this is an important adaptation of SCOPE. The algorithm is presented for general regularizations, in terms of the proximal operator. I would have liked another experiment with a different penalty. Maybe the group lasso is more challenging for distributed learning? That being said, the experimental results are very good (but please improve the plots).